# Novel Nipple Reinnervation Technique Using N. Suralis Graft

**DOI:** 10.3390/medicina60091533

**Published:** 2024-09-20

**Authors:** Jānis Lapiņš, Beatriz Soares Domingues Polita, Linda Kalniņa, Michal Grucki, Dzintars Ozols, Ansis Ģīlis, Arvīds Irmejs, Jānis Gardovskis, Jeļena Maksimenko

**Affiliations:** 1Department of Doctoral Studies, Rīga Stradiņš University, LV-1007 Riga, Latvia; linda.m.kalnina@gmail.com (L.K.); dz.ozols@gmail.com (D.O.); 2Pauls Stradiņš Clinical University Hospital, LV-1002 Riga, Latviaarvids.irmejs@rsu.lv (A.I.); jelena.maksimenko@rsu.lv (J.M.); 3Faculty of Medicine, Rīga Stradiņš University, LV-1007 Riga, Latvia; beatrizpolita@hotmail.com (B.S.D.P.);; 4Department of Pediatric Surgery, Children Clinical University Hospital, LV-1004 Riga, Latvia; 5Institute of Oncolgy and Molecular Genetics, Rīga Stradiņš University, LV-1007 Riga, Latvia; 6Department of Surgery, Rīga Stradiņš University, LV-1007 Riga, Latvia

**Keywords:** nipple reinnervation, nerve autograft, nipple-sparing mastectomy

## Abstract

Following nipple-sparing mastectomy (NSM), patients commonly experience significant impairment or total loss of nipple sensitivity, which negatively impacts the patients’ quality of life, whereas patients who retain nipple sensation postoperatively experience enhanced physical, psychosocial, and sexual well-being. Reinnervation techniques such as nerve allografting have been utilized to retain sensation. Despite the benefits of nerve allografts, such as lack of donor site morbidity, ease of use, and potentially shorter surgery time, there are shortcomings, such as the cost of commercially available acellular nerve allografts, and, most importantly, decreased sensory and motor function recovery for acellular nerve allografts with a diameter greater than 3 mm or a length greater than 50 mm. We present a technique where we performed immediate implant-based breast reconstruction combined with nipple–areola complex reinnervation using an autologous nerve graft. Following the procedure, the patient had improved sensory outcomes in the reconstructed breast and good quality-of-life indices. This report highlights the potential for sural nerve autografts in restoring breast sensation following mastectomy.

## 1. Introduction

The landscape of breast surgery has undergone a remarkable evolution in the last decade, with the increase in nipple-sparing mastectomy (NSM) rates on account of preoperative magnetic resonance imaging (MRI) and genetic testing. The scope of NSM indications has broadened, including risk-reducing strategies and therapeutic procedures [1]. The central propellers of NSM popularity include the empirical evidence affirming their surgical and oncological safety and patients’ satisfaction and quality-of-life (QoL) results after this procedure [2]. An example of this shift can be found in the Surveillance, Epidemiology, and End Results (SEER) database, where a notable escalation in NSM rates is reported, surging from 266 in 2004–2009 to 5370 in 2010–2016 [3]. Following NSM, patients commonly experience significant impairment or total loss of nipple sensitivity, which negatively impacts the patients’ quality of life (QoL), whereas patients who retain nipple sensation postoperatively experience enhanced physical, psychosocial, and sexual well-being [4]. In the preceding literature, methods of nipple reinnervation with acellular nerve allografts following post-implant-based reconstruction have been described [5,6].

This article describes a novel surgical approach that involves using an autologous nerve graft sourced from the patient’s sural nerve.

## 2. Case Presentation

We present a case of a 35-year-old female patient diagnosed with multifocal breast cancer, clinical stage IA, cT1mN0M0G3, of the right mammary gland. According to the breast ultrasound and contrast-enhanced MRI, the patient presented with three breast masses. The first lesion, in the right breast, was located slightly above the nipple medially, measuring 1.3 × 0.9 cm. The second lesion was found slightly more caudally and laterally, while a third, smaller lesion, measuring 0.5 cm, was located between the two previously described lesions. The clinicopathological characteristics of both tumors revealed the diagnosis of poorly differentiated, metaplastic/matrix-producing carcinoma (G3), with ER-0%, PR-0%, HER-2-0, and Ki-67-90% from the first tumor and poorly differentiated invasive ductal carcinoma (G3), with ER-0%, PR-0%, HER-2-0, and Ki-67-75% from the second tumor. The cytology of the right axillary lymph nodes was negative. The genetic testing revealed a BRCA1 founder (BRCA1(NM_007300.4):c.5329dup) pathogenic variant carrier. The patient underwent neoadjuvant chemotherapy (4× AC -> carboplatin/paclitaxel) with eight doses of immunotherapy (200 mg of pembrolizumab) and nine doses of immunotherapy in an adjuvant setting (200 mg of pembrolizumab) and achieved complete radiological remission.

The patient underwent a therapeutic mastectomy of the right breast and immediate reconstruction with a smooth round implant placed under the pectoralis major muscle and additional coverage of the acellular dermal matrix, as well as prophylactic mastectomy of the left breast, nipple–areola complex reinnervation with a sural nerve graft, and volume and shape reconstruction with a smooth round implant and ADM. After examining the surgical specimen, the patient was confirmed to have achieved complete pathological remission. Adjuvant radiation therapy was administered to the right-side mastectomy scar and axillary lymph nodes, with a single dose of 2.67 Gy delivered, totaling 40.05 Gy for the entire course. The patient received 9 adjuvant doses of 200 mg of pembrolizumab, completing a total of 17 doses.

The surgery was performed under general anesthesia. Both breasts and the left leg were prepared and draped. The surgery was performed by two teams: a breast surgeon and a team of two plastic surgeons with backgrounds in microsurgery.

While the breast surgeon performed the right-breast NSM, the team of plastic surgeons made the lateral incision in the inframammary crease on the left side (prophylactic side). Careful dissection along the lateral border of the pectoralis major muscle border was performed, exposing the fourth and the fifth lateral intercostal cutaneous nerves. The fourth intercostal nerve path toward the nipple was noted, while the fifth nerve pathway was more caudal. The nerves were dissected to their maximal length, carefully preserving the vasa vasorum, and then transected and securely tucked away at the lateral border of the breast. The breast surgeon switched to the left side to complete the NSM.

The sural nerve graft was harvested classically. The total length of the nerve graft was 14 cm, and the distal portion had three terminal branches.

After completing the right-side mastectomy, the volume reconstruction was performed using a smooth round implant (SOR-MR 255, GC Aesthetics, Dublin, Ireland) placed in the submuscular pocket and covered with an acellular dermal matrix (Native 3D, Decomed, Venice, Italy).

After the risk-reducing NSM was completed on the left side, the undersurface of the areola–nipple complex was revised under loop 4× magnification. A small portion of the suspected nerve stump in the central part of the areola was excised and sent to a confirmatory frozen section. After rapid confirmation that the specimen was a nerve tissue, reinnervation was performed. The thickest terminal branch of the sural nerve was coapted in an end-to-end fashion to the areola–complex nerve stump using 8-0 Ethilon (Ethicon US, LLC, Raritan, NJ, USA) epineural sutures. The smaller branches were sutured directly to the undersurface of the areola (Figure 1). The distal anastomosis was secured with a human fibrinogen/thrombin sponge (Tachosil^®^). The implant (SOR-MR 255, GC Aesthetics, Dublin, Ireland) was placed in the submuscular pocket; the lateral part was covered with an acellular dermal matrix (Native 3D, Decomed, Venice, Italy). Finally, the proximal nerve anastomosis was completed to the fourth intercostal nerve end (Figure 2), and the anastomosis was wrapped in a Tachosil^®^ sponge. The nipple–areolar complex reinnervation added 30 min to the total surgical time. Bilateral subcutaneous drains (Blake, No 15; Ethicon US, LLC) were placed in both breasts. 

To evaluate the efficacy of the nipple reinnervation with the sural nerve autograft, we performed a Semmes–Weinstein monofilament test to quantify sensation before the procedure, one month, three, and six months postoperatively. The sensory test was conducted at 9 predefined points of the breast (Figure 3): the quadrants of the areola and periareolar area, and the nipple. The sensation was quantified by target force in grams by applying the monofilament evaluator (Performance Health Supply, Inc. Cedarburg, WI, USA) and categorized as normal, diminished light touch, diminished protective sensation, loss of protective sensation, or loss of sensation/deep pressure sensation only (these numbers and evaluations were cited from ‘Touch-Test Sensory Evaluator Instructions’^®^ 2011 North Coast Medical, Inc., Morgan Hill, CA, USA) (Table 1).

## 3. Results

Bilateral mastectomy and immediate breast reconstruction with a breast implant and acellular dermal matrix were performed. Additional nipple–areola complex neurotization was performed on the left breast using an autologous nerve graft. The length of the nerve graft used for tension-free anastomosis was 7 cm.

There were no operative or postoperative complications.

The first sensory test was performed one month postoperatively. The sensation threshold was better in the left breast’s upper and lower outer quadrants (0.07 g and 1.0 g, compared with the contralateral side at 8.0 g and 4.0 g) and upper lateral areola and lower lateral areola (0.07 g and 0.16 g, compared with the contralateral side at 8 g and 0.6 g). The second test was performed three months postoperatively. The lower outer quadrant and the nipple had better results on the left breast (0.4 g and 2.0 g, compared with 1.4 g and 4.0 g). The third test, six months postoperatively, revealed improved nipple sensation compared with the previous measurement (0.008 g compared with 2 g) and better sensation compared with the contralateral nipple (0.008 g compared with 0.6 g) (Figure 4).

The preoperative breast Q-score of 58 indicated a moderate level of satisfaction with her breasts before the surgery. There was a slight improvement in satisfaction post-reconstruction (65), indicating that the patient felt somewhat better about their breasts after surgery.

## 4. Discussion

This study described the reinnervation of the nipple–areola complex using an autologous nerve graft. There was a positive difference postoperatively over six months compared with the contralateral side, where nerve reconstruction was not performed.

Several studies have found relatively low rates of sensation preservation following nipple-sparing mastectomy (NSM). Sensory recovery following NSM remains poorly understood, with authors reporting postoperative rates of a skin flap or nipple sensation of up to 47% [7,8]. Reinnervation techniques using nerve allografts have been described. Djohan et al. described the sensate implant-based breast reconstruction using a processed nerve allograft as an interposition graft connecting the donor nerve to the targeted nipple-areola complex. Their study presented sensory outcomes from 15 mastectomies in eight patients using a pressure-specified sensory device to assess sensation. They found overall improvements in mastectomy skin and nipple–areola complex sensation over time [6]. Peled et al. described nipple–areola complex neurotization utilizing a nerve allograft coapted from preserved T4 or T5 lateral intercostal nerves to subareolar nerves identified at the completion of the mastectomy. Of the 16 patients, nipple–areola complex two-point discrimination was preserved in 20 breasts (87%) [5].

The total nerve graft length necessary for tension-free grafting was 7 cm for this patient. The use of an autologous nerve graft potentially provides better reinnervation compared with allografts, thanks to revascularization via inosculation, leading to a shorter ischemia time [9]. The use of an autologous nerve graft adds no extra cost to the procedure, except for the donor site morbidity.

Since the publication of data by Millesi et al. [10], nerve autografts have been considered the standard treatment for peripheral nerve defects. Understanding the morbidity of sural nerve harvest is essential when counseling patients regarding nerve grafts. Bamba et al. conducted a systematic review on donor site morbidity after sural nerve grafting. The existing data consist of small studies with varying degrees of follow-up and a wide range of reported donor site outcomes. They found five studies reporting surface areas of sensory loss, and this generally decreased over time after sural nerve grafting. Overall, 87.2% of patients (*n* = 190) reported sensory loss, 25.6% (*n* = 42) of patients reported pain, 22.2% (*n* = 28) of patients reported cold sensitivity, and 10% (*n* = 20) of patients reported functional impairment at follow-up [11]. To evaluate the long-term outcomes of donor site morbidity, Kaoru et al. investigated problems with lower limbs in 13 patients who underwent nerve autograft with sural nerve harvesting over 15 years previously. They concluded that although donor site morbidity after sural nerve graft harvesting persisted for a long time after surgery, the foot symptoms and functional impairment were mild [12].

We performed the nerve graft harvesting parallel to the mastectomy; the extra time spent for reinnervation was 30 min. We measured the breast sensibility before the surgery, one month, three months, and six months after. The time for reinnervation to occur could be too short, so we will continue to evaluate sensibility twelve months after the procedure. The preoperative breast Q-score of 58 indicated a moderate level of satisfaction with her breasts before the surgery. There was a slight improvement in satisfaction post-reconstruction (65), indicating that the patient felt somewhat better about their breasts after surgery. 

The positive outcome of improved NAC sensation for this patient encourages us to enroll more patients in this procedure and conduct a study with statistical analysis.

## 5. Conclusions

This is the first report of this technique where immediate implant-based breast reconstruction is combined with nipple–areola complex reinnervation using an autologous nerve graft. There was a tendency for sensation restoration of the nipple–areola complex using an autologous nerve graft, and nipple sensation returned to normal after six months postoperatively.

We will continue to follow this patient and enroll more patients in this procedure.

## Figures and Tables

**Figure 1 medicina-60-01533-f001:**
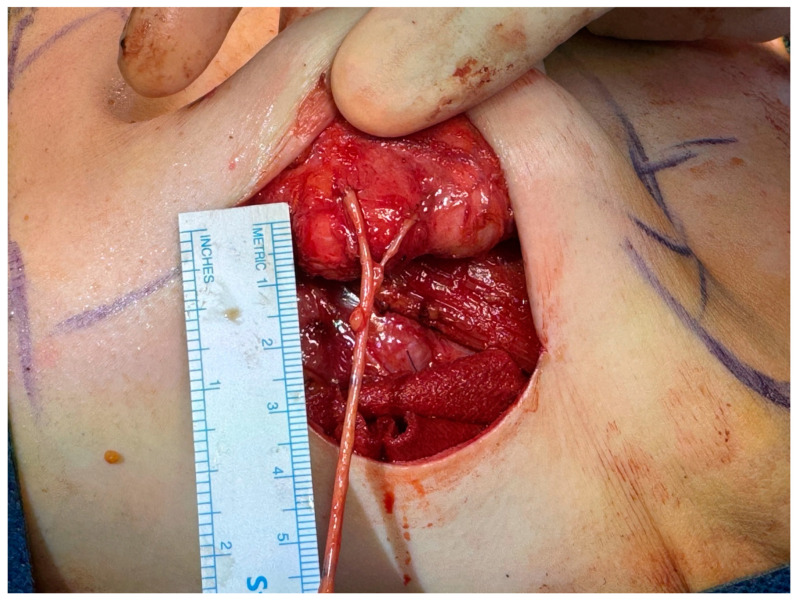
The thickest terminal branch of the sural nerve is coapted to the areola-complex nerve stump. The smaller branches sutured directly to the undersurface of the areola.

**Figure 2 medicina-60-01533-f002:**
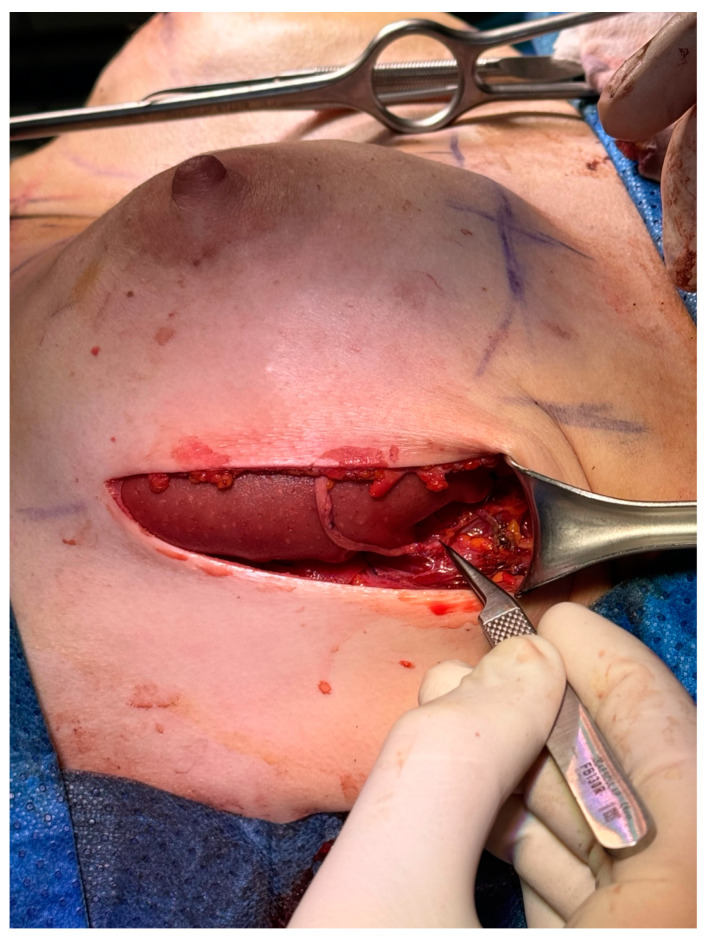
The proximal nerve anastomosis was completed to the fourth intercostal nerve end.

**Figure 3 medicina-60-01533-f003:**
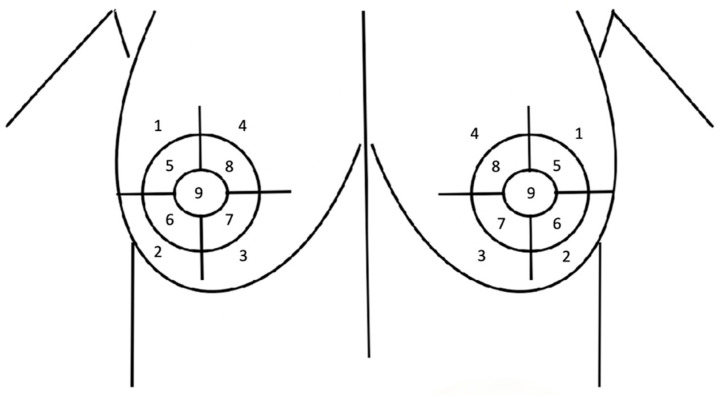
Nine predefined points of the breast: 1—upper outer quadrant (UOQ); 2—lower outer quadrant (LOQ); 3—lower inner quadrant (LIQ); 4—upper inner quadrant (UIQ); 5—areola upper lateral (AUL); 6—areola lower lateral (ALL); 7—areola lower medial (ALM); 8—areola upper medial (AUM); 9—nipple (NIP).

**Figure 4 medicina-60-01533-f004:**
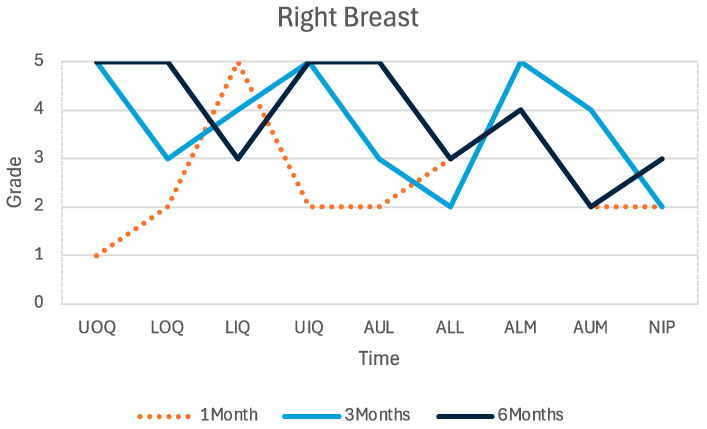
Sensibility grade of each predefined point over time.

**Table 1 medicina-60-01533-t001:** The sensation quantification by target force in grams by applying the monofilament evaluator is categorized as normal, diminished light touch, diminished protective sensation, loss of protective sensation, or loss of sensation/deep pressure sensation only.

Grade	Monofilament Size	Target Force (gm)	Interpretation
5	1.65–2.83	0.008–0.07	Normal
4	3.22–3.61	0.16–0.4	Diminished light touch
3	3.84–4.31	0.6–2	Diminished protective sensation
2	4.56–4.93	4–8	Loss of protective sensation
1	5.05–6.45	10–180	Loss of protective sensation
0	6.65	300	Loss of sensation/deep pressure sensation only

## Data Availability

The data is unavailable due to privacy/ethical reasons.

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
