# Peer review of "Novel Nipple Reinnervation Technique Using N. Suralis Graft"

_medicina, 2024, doi:10.3390/medicina60091533_

Round 1

Reviewer 1 Report

Comments and Suggestions for Authors

Although cosmetically impressive results are achieved with the nipple-sparing mastectomy technique in the treatment of breast cancer, sensory loss is detected in many studies and this situation has negative consequences in the social and sexual lives of the patients.

Nipple Reinnervation Techniques are applied to reduce this sensory loss and acellular nerve allografts are frequently used, but sensory loss is frequently encountered. The autograft applied by the authors seems to have reduced the ischemia time of the transplanted nerve and made it more effective. For the first time in the literature, a method that is both low-cost and highly effective has been developed with autograft and I think the study will contribute to the literature.

When I consider the sections of the study separately;

The abstract of the study explains the purpose of the study well and the contribution of the case to the literature is clearly stated.

The first two paragraphs in the introduction section of the study are sufficient, and it is recommended that the sections where information is provided with the case are combined with the surgical technique and results section under the title of case presentation.

I suggest that it be stated why the number of neoadjuvant-adjuvant pembrolizumab courses of the case was 10. Was the full dose not given due to toxicity? (17 doses were given in the Keynote-522 study, 8 neoadjuvant and 9 adjuvant)

The study is well supported with figures.

In the discussion section, a good comparison with the literature is made, and it is explained why autograft should be applied.

It is thought that too many paragraphs are used, especially in the discussion section, and it is recommended that the paragraphs be combined appropriately.

Best regards

Author Response

Comments 1: [I suggest that it be stated why the number of neoadjuvant-adjuvant pembrolizumab courses of the case was 10. Was the full dose not given due to toxicity? (17 doses were given in the Keynote-522 study, 8 neoadjuvant and 9 adjuvant)]
Response 1: [The patient underwent neoadjuvant chemotherapy (4x AC -> Carboplatin/Paclitaxel) with eight doses of immunotherapy (Pembrolizumab 200mg)) and nine doses of immunotherapy in an adjuvant setting (Pembrolizumab 200mg). At the time of manuscript preparation, the entire course of chemotherapy was not finished yet.] Thank you for pointing this out. We agree with this comment. Therefore, we updated the text in lines 92-95 and 104-105.
Comments 2: [The first two paragraphs in the introduction section of the study are sufficient, and it is recommended that the sections where information is provided with the case are combined with the surgical technique and results section under the title of case presentation.]
Response 2: Thank you for pointing this out. We agree with this comment. Therefore, we combined the case information with surgical technique (the text in lines 45-145).
Comments 3: [It is thought that too many paragraphs are used, especially in the discussion section, and it is recommended that the paragraphs be combined appropriately.]
Response 3: Thank you for pointing this out. We agree with this comment. Therefore, we combined the paragraphs in the discussion part (the text in lines 184-230).

Reviewer 2 Report

Comments and Suggestions for Authors

Dear Authors,

Thank you for submitting your manuscript, “Novel Nipple Reinnervation Technique using N. Suralis Graft,” to Medicina.

Your study addresses a very intriguing and important topic, offering a technique that has the potential to improve the quality of life for some patients affected by breast cancer, particularly those who have undergone mastectomy and breast reconstruction. I believe the study is well-conducted and provides valuable insights. However, I have a few questions and suggestions for further clarification:

·         The manuscript describes a case report involving a patient who underwent bilateral mastectomy with immediate breast reconstruction, but two different techniques were used: an implant placed under the pectoralis muscle for the right breast and pre-pectoral reconstruction for the left breast. Could the authors clarify the rationale behind selecting two different reconstruction methods for the same patient?

·         The clinical background seems somewhat incomplete. I suggest providing a more detailed description of the tumor's presentation, including the size of both foci, their location within the breast quadrants, whether they were palpable, and any relevant patient comorbidities and medical history.

·         The manuscript mentions that the patient was in pathological remission prior to surgery. This could be better outlined in a dedicated paragraph, following the description of the surgical treatment.

·         While I appreciate the use of the Breast-Q questionnaire pre- and post-surgery, the instrument prompts patients to reflect on the breast they are least satisfied with. In this case, there is only a slight improvement in overall breast satisfaction, and it is difficult to directly attribute this to the nipple reinnervation. It would have been interesting to administer the Breast-Q separately for both the left and right breasts, pre- and post-surgery, to gain more nuanced insight.

·         In Table 2, spontaneous reinnervation is noted for the right breast. The left breast showed greater improvement in nipple innervation, but the mean sensitivity of the other quadrants is also higher than on the right side. Could the authors offer an explanation for this? Is it possible that the less invasive surgical technique used for the left breast, given its role as a risk-reducing mastectomy, may account for this difference?

Thank you once again for the opportunity to review your work. I hope you find these comments helpful and constructive.

Sincerely,

Comments on the Quality of English Language

There are minor grammar errors throughout the manuscript. I recommend having a native English speaker or professional editing service review your manuscript for grammar and syntax. This will help refine the overall writing

Author Response

Comments 1: [The manuscript describes a case report involving a patient who underwent bilateral mastectomy with immediate breast reconstruction, but two different techniques were used: an implant placed under the pectoralis muscle for the right breast and pre-pectoral reconstruction for the left breast. Could the authors clarify the rationale behind selecting two different reconstruction methods for the same patient?]
Response 1: [Both implants were placed in the submuscular pocket as it is described in the surgical technique.]
Comments 2:  [The clinical background seems somewhat incomplete. I suggest providing a more detailed description of the tumor's presentation, including the size of both foci, their location within the breast quadrants, whether they were palpable, and any relevant patient comorbidities and medical history.]
Response 2: [According to the breast ultrasound and contrast-enhanced MRI, the patient presented with three breast masses. The first lesion, in the right breast, was located slightly above the nipple medially, measuring 1.3x0.9 cm. The second lesion was found slightly more caudally and laterally, while a third, smaller lesion, measuring 0.5 cm, was located between the two previously described lesions.] Thank you for pointing this out. We agree with this comment. Therefore, we updated the text in lines 81-86.
Comments 3: [The manuscript mentions that the patient was in pathological remission prior to surgery. This could be better outlined in a dedicated paragraph, following the description of the surgical treatment.
Response 3: [The patient underwent therapeutic mastectomy of the right breast and immediate reconstruction with a smooth round implant placed under the pectoralis major muscle and additional coverage of acellular dermal matrix, as well as prophylactic mastectomy of the left breast, nipple-areola complex re-innervation with the sural nerve graft and volume and shape reconstruction with a smooth round implant and ADM. After examining the surgical specimen, the patient was confirmed to have achieved a complete pathological remission"] Thank you for pointing this out. We agree with this comment. Therefore, we updated the text in lines 100-102.
Comments 4:   [While I appreciate the use of the Breast-Q questionnaire pre- and post-surgery, the instrument prompts patients to reflect on the breast they are least satisfied with. In this case, there is only a slight improvement in overall breast satisfaction, and it is difficult to directly attribute this to the nipple re-innervation. It would have been interesting to administer the Breast-Q separately for both the left and right breasts, pre- and post-surgery, to gain more nuanced insight.]
Response 4: Thank you for a very interesting suggestion. We will discuss this advice for future research!
Comments 5: [In Table 2, spontaneous re-innervation is noted for the right breast. The left breast showed greater improvement in nipple innervation, but the mean sensitivity of the other quadrants is also higher than on the right side. Could the authors offer an explanation for this? Is it possible that the less invasive surgical technique used for the left breast, given its role as a risk-reducing mastectomy, may account for this difference?]
Response 5: [We are assuming that a less invasive procedure could improve the sensitivity of the left breast overall. We have performed bilateral nerve reconstruction for the next patient with a similar diagnosis, and future examination of breast sensitivity could give us more answers to this question.]

Reviewer 3 Report

Comments and Suggestions for Authors

The manuscript presented for review is a proposed novel surgical technique for reinnervation of areola complex after nipple-sparing-mastectomy by using sural autologous nerve graft. The sugical technique per se is interesting and may in fact promote the quality of life in patients with breast cancer. 

My suggestions for improvement are as follows:

1. Minor English proofing needed

2. Reformate references - numbers are doubled.

3. Additional information of  Semmes-Weinstein monofilament test should be included - explain the meaning and measuring scale. With no context is dificult to evaluate the results.

4. Similar additional information needs to be included on breast-Q score.

5. Table 1 is very difficult to understand (for non-specialists the data presented has absolutely no context). Please clarify your results in a manner more reader friendly.

6. please summarize the current re-innervation techniques available in present

7. Was this procedure approved by the ethical board of the hospital? 

8. Please give details about the oncologic outcome of the patient thus fare (including radiotherapy needs).

Comments on the Quality of English Language

Minor English proofing needed.

Author Response

Comments 1: [Additional information of  Semmes-Weinstein monofilament test should be included - explain the meaning and measuring scale. With no context is dificult to evaluate the results.] and [Table 1 is very difficult to understand (for non-specialists the data presented has absolutely no context). Please clarify your results in a manner more reader friendly.]
Response 1: [The Semmes-Weinstein Monofilament test is used to evaluate touch pressure sensitivity by applying different levels of force with monofilaments of varying size. Smaller monofilaments correspond to normal sensation, as a smaller force is needed for the patient to feel them. While larger monofilaments, with larger target force, indicate varying degrees of sensory loss—from diminished light touch to total loss of protective sensation. Table 1 clarifies the interpretation of the test.]
Comments 2:  Similar additional information needs to be included on breast-Q score.
Response 2: [The BREAST-Q is a self-administered questionnaire. All BREAST-Q scales are transformed into scores that range from 0-100, independent of the type and number of modules. The scores are computed by adding the response items together and then converting the raw sum scale score to a score from 0-100. For all BREAST-Q scales, a higher score means greater satisfaction or better QOL (depending on the scale).]
Comments 3: [please summarize the current re-innervation techniques available in present.]
Response 3:  [Current re-innervation techniques for implant-based breast reconstruction are summarised in the discussion: 
Djohan et al. described the sensate implant-based breast reconstruction using pro-cessed nerve allograft as an interposition graft connecting the donor nerve to the targeted nipple-areola complex. Their study presented sensory outcomes from 15 mastectomies in 8 patients using a pressure-specified sensory device to assess sensation. They found overall improvements in mastectomy skin and nipple-areola complex sensation over time.
Peled et al. described nipple-areola complex neurotisation utilizing nerve allo-graft coapted from preserved T4 or T5 lateral intercostal nerves to subareolar nerves identified at the completion of the mastectomy. Of the 16 patients, nipple-areola complex 2-point discrimination was preserved in 20 breasts (87%).]
Comments 4: [Was this procedure approved by the ethical board of the hospital?]
Response 4: [We have approval from the ethical boards of the Pauls Stradins University hospital and the Riga Stradins University.]
Comments 5:  [Please give details about the oncologic outcome of the patient thus fare (including radiotherapy needs).]
Response 5: [The patient underwent therapeutic mastectomy of the right breast and immediate reconstruction with a smooth round implant placed under the pectoralis major muscle and additional coverage of acellular dermal matrix, as well as prophylactic mastectomy of the left breast, nipple-areola complex reinnervation with the sural nerve graft and volume and shape reconstruction with a smooth round implant and ADM. After examining the surgical specimen, the patient was confirmed to have achieved a complete pathological remission. 
Adjuvant radiation therapy was administered to the right-side mastectomy scar and axillary lymph nodes, with a single dose of 2.67 Gy delivered, totaling 40.05 Gy for the entire course.] Thank you for pointing this out. We agree with this comment. Therefore, we updated the text in lines 102-104.

Round 2

Reviewer 2 Report

Comments and Suggestions for Authors

Dear Authors,

Thank you for submitting the revised version of your manuscript review.

I have carefully reviewed your revised manuscript. I am pleased to see that you have thoughtfully and thoroughly addressed the concerns raised. The revisions have significantly improved the clarity and overall quality of your work. I am confident that the revised version is now much stronger as a result.

I believe that your manuscript is now suitable for publication, and I appreciate the effort you put into making the necessary changes. 

Best regards

Reviewer 3 Report

Comments and Suggestions for Authors

The authors have addressed my concerns adequately and the manuscipt can be published in the current form.